# Analyses of Molecular Characteristics and Enzymatic Activities of Ovine HSD17B3

**DOI:** 10.3390/ani11102876

**Published:** 2021-09-30

**Authors:** Mohammad Sayful Islam, Junsuke Uwada, Junki Hayashi, Kei-ichiro Kikuya, Yuki Muranishi, Hiroyuki Watanabe, Kazuhide Yaegashi, Kazuya Hasegawa, Takanori Ida, Takahiro Sato, Yoshitaka Imamichi, Takeshi Kitano, Yoshimichi Miyashiro, Rafiqul Islam Khan, Satoru Takahashi, Akihiro Umezawa, Nobuo Suzuki, Toshio Sekiguchi, Takashi Yazawa

**Affiliations:** 1Department of Biochemistry, Asahikawa Medical University, Asahikawa 078-8510, Japan; sayful@asahikawa-med.ac.jp (M.S.I.); uwada@asahikawa-med.ac.jp (J.U.); a160082@ed.asahikawa-med.ac.jp (J.H.); a160029@ed.asahikawa-med.ac.jp (K.-i.K.); ph_rafiq@yahoo.com (R.I.K.); 2Department of Life and Food Science, Obihiro University of Agriculture and Veterinary Medicine, Obihiro 080-8555, Japan; muranishi@obihiro.ac.jp (Y.M.); hiwatanabe@obihiro.ac.jp (H.W.); 3Animal Lab Ltd., Asahikawa 070-8012, Japan; info@animal-lab.com; 4Faculty of Health and Medical Science, Teikyo Heisei University, Tokyo 170-8445, Japan; k.hasegawa@thu.ac.jp; 5Center for Animal Disease Control, Faculty of Medicine, University of Miyazaki, Miyazaki 889-1692, Japan; a0d203u@cc.miyazaki-u.ac.jp; 6Division of Molecular Genetics, Institute of Life Sciences, Kurume University, Fukuoka 830-0011, Japan; satou_takahiro@kurume-u.ac.jp; 7Department of Marine Bioscience, Faculty of Marine Bioscience, Fukui Prefectural University, Fukui 917-0003, Japan; imamichi@fpu.ac.jp; 8Department of Biological Sciences, Graduate School of Science and Technology, Kumamoto University, Kumamoto 860-8555, Japan; tkitano@kumamoto-u.ac.jp; 9ASKA Pharma Medical Co., Ltd., Kawasaki 251-8555, Japan; miyashiro@ap-med.co.jp; 10Department of Pharmacy, University of Rajshahi, Rajshahi 6205, Bangladesh; 11Department of Pediatrics, Asahikawa Medical University, Asahikawa 078-8510, Japan; satoru5p@asahikawa-med.ac.jp; 12Department of Reproduction, National Research Institute for Child Health and Development, Tokyo 157-8535, Japan; umezawa@1985.jukuin.keio.ac.jp; 13Noto Marine Laboratory, Division of Marine Environmental Studies, Institute of Nature and Environmental Technology, Kanazawa University, Kanazawa 927-0553, Japan; nobuos@staff.kanazawa-u.ac.jp (N.S.); t-sekiguchi@se.kanazawa-u.ac.jp (T.S.)

**Keywords:** HSD17B3, ovine, androgen, *Ovis aries*

## Abstract

**Simple Summary:**

Mutations of 17β-hydroxysteroid dehydrogenase type3 (HSD17B3) gene cause disorder of sex differentiation (DSD). In this study, the open reading frame sequence of ovine HSD17B3 was revealed, and the effects of amino acid substitution on ovine and human HSD17B3 enzymatic activities were evaluated. Although ovine HSD17B3 has a conserved amino acid sequence, it possesses two amino acid substitutions that are consistent with the reported variants of human HSD17B3. Substitution of these amino acids in ovine HSD17B3 for those in human did not affect the enzymatic activities. Similarly, substitution of these amino acids of human HSD17B3 for those in ovine also did not affect the enzymatic activities. However, enzymatic activities declined in the missense mutations of the HSD17B3 gene associated with DSD, which occurred in the conserved amino acids between both species.

**Abstract:**

17β-hydroxysteroid dehydrogenase type 3 (HSD17B3) converts androstenedione (A4) into testosterone (T), which regulates sex steroid production. Because various mutations of the HSD17B3 gene cause disorder of sex differentiation (DSD) in multiple mammalian species, it is very important to reveal the molecular characteristics of this gene in various species. Here, we revealed the open reading frame of the ovine HSD17B3 gene. Enzymatic activities of ovine HSD17B3 and HSD17B1 for converting A4 to T were detected using ovine androgen receptor-mediated transactivation in reporter assays. Although HSD17B3 also converted estrone to estradiol, this activity was much weaker than those of HSD17B1. Although ovine HSD17B3 has an amino acid sequence that is conserved compared with other mammalian species, it possesses two amino acid substitutions that are consistent with the reported variants of human HSD17B3. Substitutions of these amino acids in ovine HSD17B3 for those in human did not affect the enzymatic activities. However, enzymatic activities declined upon missense mutations of the HSD17B3 gene associated with 46,XY DSD, affecting amino acids that are conserved between these two species. The present study provides basic information and tools to investigate the molecular mechanisms behind DSD not only in ovine, but also in various mammalian species.

## 1. Introduction

Androgens are sex steroid hormones that play various roles both in male and female physiological processes via pathways involving the androgen receptor (AR). Testosterone (T) is the most important androgen, and is produced from cholesterol in the gonads and adrenals in a series of steps by cytochrome P450 hydroxylases and hydroxysteroid dehydrogenases (HSDs) [1,2]. First, 17-HSDs catalyze the final step, the conversion of androstenedione (A4) into T. Further, 17-HSDs have been identified as the enzymes that interconvert 17-keto and 17-hydroxy sex steroids, which include at least 14 family members [3]. Moreover, 17-HSDs catalyze the reactions of not only steroid hormone metabolism, but also other metabolic pathways. Among the family members, HSD17B1 and HSD17B3 are involved in gonadal T production in mammals. 

HSD17B1 is mainly expressed in ovarian granulosa cells to regulate the production of active sex steroids [4,5]. It is also detectable in some other tissues, such as placenta, adipose tissue, and testis, in a species-specific fashion. HSD17B1 efficiently catalyzes the conversion of the weak estrogen estrone (E1) into active estradiol (E2). It can also convert A4 to T as active androgen or a precursor for E2 production catalyzed by CYP19A1.

HSD17B3 is expressed almost exclusively in testicular Leydig cells to regulate the production of active androgens [6,7]. It contributes to the development of male sexual characteristics by converting A4 into T during fetal life and puberty. Mutations of HSD17B3 genes cause a disorder of sexual development (DSD) and undermasculinizaion as a result of low T production and an abnormal A4/T ratio in multiple species, such as human, dog, and mouse [6,8,9,10,11]. In livestock, DSD represents one of the major causes of reduced productivity [12,13,14,15]. Therefore, an understanding of sex differentiation-related genes, including HSD17B3, is necessary to explore new mutations/variants and to develop diagnostic tools to identify them. It is also important to study the molecular mechanisms of sex differentiation. To date, mutations of DSD-related genes, including HSD17B3, have never been reported in sheep. However, as other animal species, the existence of 54,XY DSD sheep, despite the presence of a sex-determining region Y gene (SRY-positive), has been reported [12]. Furthermore, various studies have used prenatal sheep to investigate the effects of T on gonadal development and sexual dimorphism in the brain [16,17,18,19,20,21,22,23]. Nevertheless, the molecular characteristics of the HSD17B3 gene in sheep have remained unclear. In this study, we identified the open reading frame (ORF) of this gene and compared the enzymatic activities of HSD17B3 with those of ovine HSD17B1. We also investigated the effects of amino acid substitution of HSD17B3 genes on the associated enzymatic activities.

## 2. Materials and Methods

### 2.1. Animals and Tissue Collection

This study was conducted in accordance with the National Institutes of Health Guide for Care and Use of Laboratory Animals following protocols approved by the Obihiro University of Agriculture and Veterinary Medicine Committee on Animal Care (Permission number: 20–25). Ovine fresh placenta was collected from a sheep farm and sheep ovarian tissue was obtained from the slaughterhouse in Obihiro. Ovine liver, intestine, and ovaries were collected from immature sheep at a local abattoir (Asahikawa, Hokkaido, Japan).

### 2.2. Reverse Transcriptase-Polymerase Chain Reaction (RT-PCR) and Sequencing

Total RNA from each tissue was extracted using TRIzol (Thermo Fisher Scientific, Waltham, MA, USA) or TRIsure reagent (Bioline, Luckenwalde, Germany). Total RNA of ovine testis and adrenal was purchased from Zyagnen (San Diego, CA, USA). RT-PCR was performed as described [24,25,26]. The cDNA was synthesized from total RNA of each tissue using random hexamers and SuperScript III Reverse Transcriptase (Thermo Fisher Scientific, Waltham, MA, USA). PCR was performed using Ex Taq (Takara Bio Inc., Shiga, Japan), according to the manufacturer’s instructions. The reaction products of the RT-PCR assay were subjected to electrophoresis in a 1.25% agar gel, and the resulting bands were visualized by staining with ethidium bromide. The primers used for PCR are described in Appendix A. To identify the ORF of ovine HSD17B3, the PCR product obtained using testis cDNA as template was cloned into the pGEM-T Easy Vector (Promega Cooperation, Madison, Woods Hollow Road Madison, WI, USA) and sequenced on an ABI PRISM 3500 Genetic Analyzer with a Big-Dye Sequencing Kit version 3.1 (Thermo Fisher Scientific, Waltham, MA, USA), using universal primers (SP6 and T7).

### 2.3. Sequence Alignment and Phylogenetic Analysis

The alignment analysis of HSD17B3 sequences was performed using Clustal W. The neighbor-joining phylogenetic tree was constructed using MEGA version X. Analyzed proteins and their accession numbers are as follows: human HSD17B3 (CAG46692.1), monkey HSD17B3 (NP_001253433.1), goat HSD17B3 (XP_005684205.2), bovine HSD17B3 (AAI09701.1), porcine HSD17B3 (NP_001231719.1), rabbit HSD17B3 (XP_002708310.1), horse HSD17B3 (XP_023482861.1), donkey (XP_014712426.1), dog HSD17B3 (XP_003638918.1), cat HSD17B3 (XP_003995509.1), mouse Hsd17b3 (NP_032317.2), rat Hsd17b3 (NP_446459.1), and chicken Hsd17b3 (XP_425046.4).

### 2.4. Cell Culture, Transfection, and Luciferase Assay

Human embryonic kidney 293 (HEK293) and African green monkey kidney fibroblast-derived CV-1 cells (ATCC, Manassas, VA, USA) were cultured in DMEM supplemented with 10% fetal bovine serum (FBS) and penicillin (100 IU/mL) /streptomycin (100 μg/mL). They were transfected using HilyMax (Dojindo Laboratories, Kumamoto, Japan). One day before transfection, cells were seeded on 48-well plates and cultured with DMEM supplemented with 10% Hyclone Charcoal/Dextran treated FBS (GE Healthcare UK Ltd., Buckinghamshire, England). HEK293 cells were transfected with expression vectors of GFP and HSD17Bs. At 2 days post-transfection, they were treated with vehicle or steroids for 2 (A4) or 3 h (E1). Then, supernatants of culture media were collected for the incubation of CV-1 cells or E2 measurements. CV-1 cells were transfected with reporter and AR-expression vectors. At 24 h post-transfection, they were treated with vehicle (EtOH), A4 (1 nM), T (1 nM), or supernatant of HEK293 cell culture media for 24 h. Transfections were performed in triplicate within a single experiment. Luciferase assays were performed as described previously [27]. Each data point represents the mean of at least four independent experiments.

### 2.5. Plasmids

The pQCXIP expressing ovine HSD17B3 and HSD17B1 conjugating FLAG fusion tag in their C-terminus and N-terminus, respectively, were generated by amplifying the open reading frame of each gene, adding a fusion tag to each end, and cloning them into a pQCXIP vector (Invitrogen, Carlsbad, CA, USA). Ovine AR expression vector was generated by amplifying its open reading frame and cloning it into a pQCXIP vector. pQCXIP-green fluorescent protein (GFP), Slp-ARU/Luc reporter, and pQCXIP-human AR were prepared as described [7,28]. Constructs having mutations in HSD17B3 genes were prepared by the QuikChange Site-Directed Mutagenesis Kit (Agilent Technologies Inc., Santa Clara, CA, USA). The nucleotide sequences of the constructs were confirmed by DNA sequencing as described above.

### 2.6. Western Blotting Analyses

Extraction of total proteins from cultured cells and subsequent quantification were conducted as described previously [29]. Equal amounts of protein (50 or 100 μg) were resolved using 10% SDS-PAGE and transferred to polyvinylidene difluoride membranes. Western blot analyses of FLAG, HSD17B3, and GAPDH were performed with antibodies directed against FLAG (1:1000; M2, Sigma-Aldrich Co. LLC, Saint Louis, MO, USA), HSD17B3 (1:1000; PAF173Hu01, Cloud-Clone Corp., Katy, TX, USA), and GAPDH (1:1000; 14C10; Cell Signaling Technology, Inc., Danvers, MA, USA), respectively. Immunoreactive proteins were detected using horseradish peroxidase-labeled secondary antibodies (1:10,000; Jackson ImmunoResearch Labs, West Grove, PA, USA) with Clarity Western ECL substrate (Bio-Rad Laboratories Inc., Hercules, CA, USA). Signal intensity was calculated using Image J software (Appendix A, [30]).

### 2.7. Measurements by Liquid Chromatography-Tandem Mass Spectrometry (LC-MS/MS)

Quantification of E2 in culture media by LC-MS/MS is based on the methods as described ([27,28,31], ASKA Pharma Medical Corporation, Kanagawa, Japan). As internal standards, 17β-E2-13C-4 was added to a medium, which was diluted with distilled water. The steroids were extracted with methyl tert-butyl ether (MTBE). After the MTBE layer was evaporated to dryness, the extract was dissolved in 0.5 mL of methanol and diluted with 1 mL of distilled water. The sample was applied to an OASIS MAX cartridge, which had been successively conditioned with 3 mL of methanol and 3 mL of distilled water. After the cartridge was washed with 1 mL of distilled water, 1 mL of methanol/distilled water/acetic acid (45:55:1, *v*/*v*/*v*), and 1 mL of 1% pyridine solution, the steroids were eluted with 1 mL of methanol/pyridine (100:1, *v*/*v*). After evaporation, the residue was reacted with 50 μL of mixed solution (80 mg of 2-methyl-6-nitrobenzoic anhydride, 20 mg of 4-dimethylaminopyridine, 40 mg of picolinic acid, and 10 µl of triethylamine in 1 mL of acetonitrile) for 30 min at room temperature. After the reaction, the sample was dissolved in 0.5 mL of ethyl acetate/hexane/acetic acid (15:35:1, *v*/*v*/*v*) and the mixture was applied to an InertSep SI cartridge, which had been successively conditioned with 3 mL of acetone and 3 mL of hexane. The cartridge was washed with 1 mL of hexane and 2 mL of ethyl acetate/hexane (3:7, *v*/*v*). E2 was eluted with 2.5 mL of acetone/hexane (7:3, *v*/*v*). After evaporation, the residue was dissolved in 0.1 mL of acetonitrile/distilled water (2:3, *v*/*v*) and the solution was subjected to an LC-MS/MS.

### 2.8. Statistical Analyses

Data are presented as the mean ± SEM or the mean ± SD. Differences between groups were assessed by the one-way ANOVA followed by Tukey’s multiple comparison test using EZR (Easy R, Saitama Medical Center, Jichi Medical University, Saitama, Japan), which is a graphical user interface for R (The R Foundation for Statistical Computing, Vienna, Austria) as described [31]. *p*-Values less than 0.05 were considered significant.

## 3. Results

### 3.1. Conservation of Ovine HSD17B3 Gene

Although predicted ovine HSD17B3 is registered in a public database, there are three isoforms that have different deduced amino acid sequences. To identify the true open reading frame (ORF), ovine HSD17B3 was amplified from testis cDNA as a template by RT-PCR using primers constructed from the conserved sequence in the 5′ and 3′ non-coding regions of Bovidae HSD17B3. Ovine HSD17B3 cDNA (DDBJ accession number: LC631120) comprises an open reading frame (ORF) of 933 bp that encodes 310 amino acids (Appendix A). This length is consistent with data of XM_042243038, despite the presence of some different amino acids. It is also consistent with the length of HSD17B3 from the other mammalian species, such as bovine, goat, porcine, and human (Figure 1A). Alignment of the deduced amino acid sequences showed the presence of a well-conserved cofactor-binding motif (GXXXGXG), an active site (YXXXK), and androgen-binding amino acids.

Ovine HSD17B3 showed the highest identity with that of goat in terms of both nucleotide (98.82%) and amino acid (98.39%) sequences (Appendix A). Bovine homologs also showed high identity (nucleotide: 96.14%, amino acid: 93.61%). Ovine HSD17B3 showed moderate identity with the homologs in other mammalian species (nucleotide: 77.72–89.14%, amino acid: 71.12–83.81%), whereas its identity with chicken homologs was low (nucleotide: 66.49%, amino acid: 57.86%). A phylogram based on amino acid sequences placed ovine in a clade of Cetartiodactyla with high bootstrap values (Figure 1B). As in the case of other mammals, ovine HSD17B3 was strongly expressed in testis (Figure 1C).

### 3.2. Evaluation of Enzymatic Activities of Ovine HSD17B3

Next, the enzymatic activities of ovine HSD17B3 to produce T were evaluated using our system established for human HSD17B3, which quantifies the conversion from A4 to T based on AR-mediated transactivation [7]. Although the ovine AR gene is registered in NCBI database (https://www.ncbi.nlm.nih.gov/nuccore/NM_001308584.1, accessed on 31 August 2021), to the best of our knowledge, its responsiveness to androgens has never been investigated. To construct an ovine-based system, a potential for ovine AR-mediated transactivation between T and A4 at various concentrations was analyzed in CV-1 cells. The ovine AR-mediated transactivation was increased by T from a concentration of 10^−10^ M (Figure 2A). In contrast, A4 hardly increased AR-mediated transactivation at this concentration. Although T had higher potentials than A4 at all concentrations, the ratio of T-induced activity to A4-induced activity was extremely high at concentrations of 10^−10^ and 10^−9^ M (Figure 2B). These results suggest that, based on the efficiency of transactivation, ovine AR can also discriminate between T and A4 at these concentrations. Next, HEK293 cells were transfected with the expression vectors of GFP and ovine HSD17B3 (Figure 2C). At 2 days post-transfection, the cells were incubated with media containing A4 at 10^−9^ M until 2 h. Supernatants of culture media were collected at each time point, and added to CV-1 cells transfected with ARE-Luc and AR expression vectors. AR-mediated transactivation in CV-1 cells increased linearly by each culture medium in a manner dependently on the culture period until 1 h after A4 addition in HSD17B3-expressing HEK293 cells. This reflected high enzymatic activities for converting A4 to T in HSD17B3-expressing cells. In contrast, culture media from GFP-expressing cells never activated AR-mediated transactivation, even at 2 h after A4 addition (Figure 2C,D). Similarly, culture media from both GFP- and ovine HSD17B3-expressing HEK293 cells supplemented with A4 at a concentration of 10^−10^ M hardly induced AR-mediated transactivation (Figure 2D).

Using this system, we compared the enzymatic activities of ovine HSD17B3 and HSD17B1 (Figure 3). Ovine HSD17B1 also showed high enzymatic activities, although these were slightly lower (but not significant) than HSD17B3 (Figure 3B). To compare the function to produce active estrogen, we also investigated the enzymatic activities of HSD17B3 and HSD17B1 for converting E1 into E2 (Figure 3C). Although HSD17B3-expressing cells significantly converted E1 into E2 compared with GFP-expressing cells, the levels of conversion were much lower than that of HSD17B1-expressing cells.

### 3.3. Enzymatic Activities Associated with Various Missense Mutations in Ovine and Human HSD17B3 Genes

Although various missense mutations with or without the manifestation of 46,XY, DSD have been reported in the human HSD17B3 gene, the enzymatic activities of mutant proteins have often not been defined [6,32,33,34]. Ovine HSD17B3 possesses two amino acid substitutions (V31I and G289S) that are consistent with the reported variants of human HSD17B3 (Figure 4A,B). We expressed these amino acid-substituted enzymes in HEK293 cells, and compared the enzymatic activities with that of wild-type protein using the above-mentioned system. As shown in Figure 4C, the levels of ovine AR-mediated transactivation by culture media of I31V- and S289G-expressing cells were comparable to that of ovine wild-type protein-expressing cells. Similarly, substitutions of amino acids at the same positions in human HSD17B3 (V31I and G289S) hardly affected human AR-mediated transactivation by culture media (Figure 4D). In contrast, mutations in amino acids conserved between these two species (L128S and P193H) decreased AR-mediated transactivation by culture media.

## 4. Discussion

Mammalian HSD17B3 genes possess highly conserved sequences to exert common functions to convert A4 into T in testicular Leydig cells [6]. Although various previous studies investigated the effects of T on the gonadal development and sexual dimorphism of the brain in prenatal sheep [16,17,18,19,20,21,22,23], the molecular characteristics of this gene have not been revealed. In this study, we identified the ORF of ovine HSD17B3 and evaluated its enzymatic activity. The amino acid sequence of ovine HSD17B3 showed high similarity to that of Bovidae homologs, whereas it showed moderate similarity to other mammalian homologs, including human HSD17B3. However, important amino acids, such as those forming the cofactor-binding motif, active site, and androgen-binding region, were almost completely identical in all species. This conservation is likely to have led to the finding that A4 added to culture media at 10^−9^ M was mostly converted into T within 2 h in cells expressing HSD17B3 derived from various species [7,31]. Using this characteristic, we established a system for measuring the enzymatic activities of HSD17B3 via ovine AR-mediated transactivation, based on our previous study using human AR [7]. Although this system is an indirect method to quantify T production, there are some advantages. It is very sensitive and rapidly detects the conversion of substrates, due to a low background activity of C3 group nuclear receptors and a good response to T by ectopic expression of AR in CV-1 cells. In addition, it can be achieved using only common cultured cells (HEK293 cells and CV-1 cells). Ovine AR-mediated reporter assay is possibly useful not only for evaluating enzymatic activities of HSD17B3 mutants, but also for evaluating transcriptional activities of AR mutants. Mutations of the AR gene are also the cause of DSD in other animal species [35], although gene mutations causing DSD, including HSD17B3 and AR, have never been reported in sheep.

Mutation of human HSD17B3 causes 46,XY DSD as a result of low T concentration. Newborn males with 17β-HSD3 deficiency have complete or predominantly female external genitalia with a blind vaginal pouch [6,8,9]. To date, more than 50 pathogenic and benign mutations in the HSD17B3 gene have been reported. In addition to humans, it was recently reported in DSD dogs that truncation of the encoded protein was caused by 2-bp deletion of the HSD17B3 gene [10]. Furthermore, male Hsd17b3KO mice were reported to show a delay in puberty, owing to an abnormal A4/T ratio [11]. These findings indicate that the HSD17B3 gene plays important roles in sex differentiation in various mammalian species. Therefore, it is conceivable that mutations of HSD17B3 could be the cause of DSD in sheep. Because XY DSD was also reported in sheep [12], it is expected that the sequence information and measurement of enzymatic activity of HSD17B3 in this study can contribute to revealing the mechanisms of sex differentiation and the causes of DSD in sheep.

Consistent with a previous report of human HSD17B3, ovine HSD17B3 weakly converts E1 into E2 [36]. Although the physiological significance of this activity is unclear, HSD17B3 may contribute to testicular E2 production. In support of this hypothesis, testicular E2 levels in Hsd17b3 KO mice were found to be decreased, despite E1 levels being increased [11]. Estrogen-estrogen receptor (ER) pathways are essential for spermatogenesis [37,38,39]. Male Cyp19a1 and ERα KO mice are infertile as a result of the defective spermatogenesis [40,41]. In sheep, it has often been reported that active spermatogenesis is concomitant with the elevation of plasma E2 levels [42,43]. Therefore, it would be interesting to investigate the function of HSD17B3 in testicular E2 production and spermatogenesis.

Consistent with the findings in other species, ovine HSD17B1 is strongly expressed in the ovary (our unpublished data). In addition to the conversion of E1 into E2, ovine HSD17B1 can strongly convert A4 to T. Previous studies showed that murine Hsd17b1 equally converts E1 and A4 into E2 and T, whereas the catalytic efficiency of human HSD17B1 for the A4 to T reaction is very weak [44]. However, using the AR-mediated transactivation system, we suggested that these results could be caused by substrate inhibition of human HSD17B1 through unphysiological A4 concentrations [7]. Human HSD17B1 produces T from A4 at a physiological concentration (10^−9^ M), a level similar to those of mouse Hsd17b1 and porcine HSD17B1 [7]. Therefore, it is conceivable that mammalian HS17B1 can efficiently convert not only E1 to E2, but also A4 to T.

Although the amino acid sequences of HSD17B3 proteins are relatively conserved in mammals, some amino acid substitutions that are consistent with the reported variants of human homolog have occurred in other animal species. Ovine HSD17B3 possesses two such amino acid substitutions (V31I and G289S, Figure 4B). Because substitutions of amino acids at these positions in human and ovine HSD17B3 did not affect the enzymatic activities, it is reasonable that V31I and G289S substitutions are benign variants. In addition to DSD, it was reported that G289S is associated with the risks of prostate cancer and hypospadias [33,45]. However, consistent with our results, the enzymatic activity of these mutant proteins associated with this mutation was reported to be similar to that of wild-type protein in multiple previous studies [7,46,47]. Furthermore, the amino acid residue at this position is serine in almost all mammalian species except for primates (Figure 1A), although no studies have shown the associated susceptibility to prostate cancer and hypospadias in particular animal species. Hence, it is conceivable that G289S is not involved in these diseases. In contrast, missense mutations of the HSD17B3 gene in 46XY, DSD patients (L128S and P193H) result in a loss of enzymatic activities (Figure 4D). These mutations affect amino acids that are conserved between ovine and human. Because HSD17B3 genes play the conserved roles by converting A4 into T at least in male mammals, more multi-species sequence comparisons should provide valuable information for analyzing the relationship of gene mutations and variants with DSD and other diseases in various species.

## 5. Conclusions

We identified the correct ORF of the ovine HSD17B3 gene. Although ovine HSD17B3 has an amino acid sequence is conserved relative to those of other mammalian species, it possesses two amino acid substitutions that are consistent with the reported variants of human HSD17B3. Substitutions of these amino acids in ovine HSD17B3 to those in the human one did not affect the enzymatic activities. However, the enzymatic activities decreased in the missense mutations of the HSD17B3 gene associated with 46,XY DSD, affecting amino acids that are conserved between both species. Taking these findings together, the present study provides valuable insights for investigating the molecular basis of DSD not only in ovine, but also in other species.

## Figures and Tables

**Figure 1 animals-11-02876-f001:**
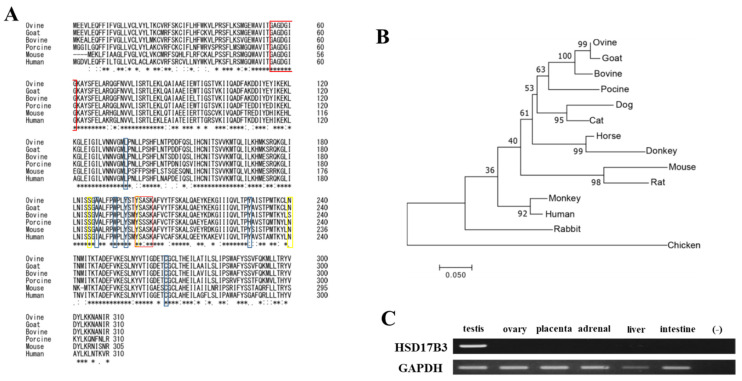
Alignment (**A**), phylogenic analyses (**B**), and tissue expression for ovine HSD17B3. (**A**) Alignment of the deduced HSD17B3 amino acid sequences of ovine, goat, bovine, porcine, mouse, and human. The asterix (*) are the conserved amino acids among all species. Conserved cofactor-binding motif (GXXXGXG) and active site (YXXXK) are shown by redline boxes. Putative amino acids for strong and weak androgen-binding are shown by the yellow line and blue line box, respectively. (**B**) The phylogenetic tree of HSD17B3 deduced amino acids. Bootstrap values (100 resamplings) are indicated by numbers. (**C**) Tissue expression of HSD17B3 genes. RT-PCR analyses of each gene in ovine testis, ovary, placenta, adrenal, liver, and intestine.

**Figure 2 animals-11-02876-f002:**
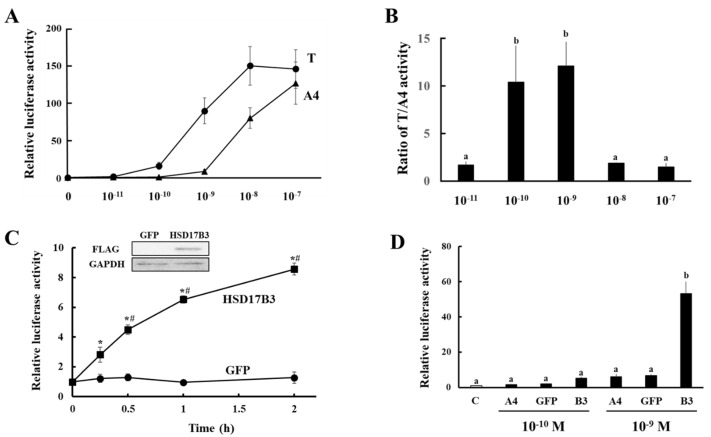
Evaluation of enzymatic activities of HSD17B3 by ovine androgen receptor (AR)-mediated transactivation in reporter assays. (**A**) Ovine AR-mediated transactivation by A4 and T. CV-1 cells were transfected with the ARE-Luc vector and the ovine AR-expression vector. At 24 h post-transfection, cells were incubated with or without increasing concentrations of each androgen for 24 h. Data represent the mean ± SEM of at four independent experiments. (**B**) Comparison of the potentials of ovine AR-mediated transactivation between A4 and T at each concentration. Values of A4 were defined as 1. (**C**) Incubation time-dependent activation of ovine AR-mediated transcription by culture media from GFP- or HSD17B3-introduced HEK293 cells. CV-1 cells were transfected with ARE-Luc and ovine AR-expression vectors. At 24 h post-transfection, cells were incubated with each culture medium collected from HEK293 cells transfected with GFP or HSD17B3 expression vector at the indicated time after A4 (1 nM) addition for 24 h. Values for A4 (1 nM) addition in ARE-Luc vector- and ovine AR-expression vector-transfected CV-1 cells were defined as 1. Data represent the mean ± SEM of four independent experiments (* *p* < 0.05 vs. 0 h, # *p* < 0.05 vs. GFP group at the corresponding time). The left inset shows the Western blot analyses that were performed with the antibodies against FLAG-tag and GAPDH using lysates derived from GFP- or FLAG-tagged ovine HSD17B3-introduced HEK293 cells. (**D**) Evaluation of the enzymatic activities of ovine HSD17B3. Activation of ovine AR-mediated transcription by culture media from GFP- or ovine HSD17B3-introduced HEK293 cells. CV-1 cells were transfected with ARE-Luc and ovine AR-expression vectors. At 24 h post-transfection, cells were incubated with vehicle (**C**), A4 (10^−10^ or 10^−9^ M) and each gene-expressing HEK293 cell’s culture medium collected at 2 h after A4 addition for 24 h. Values of the vehicle were defined as 1. Data represent the mean ± SEM of four independent experiments. Values marked by the different letters are significantly different with each other (*p* < 0.05).

**Figure 3 animals-11-02876-f003:**
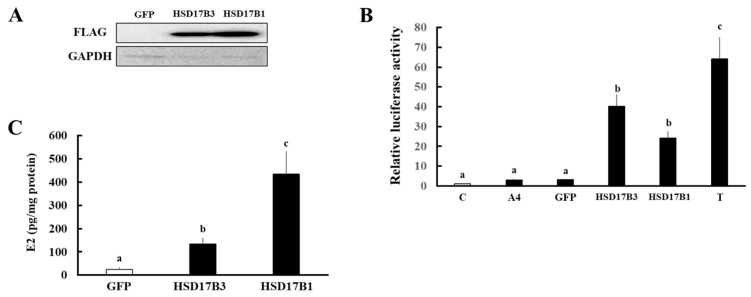
Evaluation of the enzymatic activities of ovine HSD17B3 and HSD17B1 using culture media from each gene-transfected HEK293 cells. (**A**) Western blot analyses were performed with the antibodies against FLLAG-tag and GAPDH using lysates of GFP-, FLAG-tagged ovine HSD17B3-, and FLAG-tagged ovine HSD17B1-introduced HEK293 cells. (**B**) Activation of ovine AR-mediated transcription by HSD17B3 and HSD17B1 using culture media from each gene-transfected HEK293 cell. CV-1 cells were transfected with ARE-Luc and ovine AR-expression vectors. At 24 h post-transfection, cells were incubated with vehicle (lane C), A4 (1 nM), T (1 nM), culture medium from GFP-, HSD17B3-, and HSD17B1-expressing HEK293 cells collected at 2 h after A4 addition for 24 h. Values of the vehicle were defined as 1. Data represent the mean ± SEM of four independent experiments. Values marked by the different letters are significantly different from each other (*p* < 0.05). (**C**) Concentrations of E2 in culture medium from GFP-, HSD17B3-, and HSD17B1-introduced HEK293 cells at 3 h after addition of E1 (1 nM). Each column represents the mean ± SD of three independent experiments. Values marked by the different letters are significantly different with each other (*p* < 0.05).

**Figure 4 animals-11-02876-f004:**
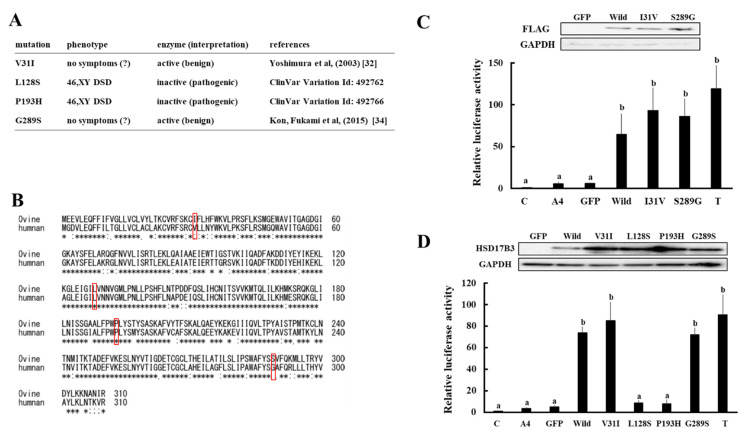
Enzymatic activities of missense mutations in ovine and human HSD17B3 genes. (**A**) Clinical symptoms and enzymatic activities of each mutation in the human HSD17B3 gene. (**B**) Comparison of the deduced HSD17B3 amino acid sequences of ovine and human. The asterix (*) are the conserved amino acids between both species. Amino acids that can mutate as shown in (**A**) are indicated by the boxes. (**C**) Upper panel shows Western blot analyses performed with the antibodies against FLAG-tag and GAPDH using lysates of GFP-, wild-type FLAG-tagged ovine HSD17B3- and FLAG-tagged each mutant HSD17B3-introduced HEK293 cells. Lower panel shows activation of ovine AR-mediated transcription by culture media from wild-type and mutant ovine HSD17B3-introduced HEK293 cells. CV-1 cells were transfected with ARE-Luc and ovine AR-expression vectors. At 24 h post-transfection, cells were incubated with vehicle (**C**), A4 (10^−9^ M), T (10^−9^ M), and each gene-expressing HEK293 cell’s culture medium collected at 2 h after A4 addition. Values of the vehicle were defined as 1. Data represent the mean ± SEM of four independent experiments. Values marked by the different letters are significantly different with each other (*p* < 0.05). (**D**) Upper panel shows Western blot analyses that were performed with the antibodies against HSD17B3 and GAPDH using lysates of GFP-, wild-type human HSD17B3-, and each mutant HSD17B3-introduced HEK293 cells. Lower panel shows activation of human AR-mediated transcription by culture media from wild-type and mutant human HSD17B3-introduced HEK293 cells. CV-1 cells were transfected with ARE-Luc and ovine AR-expression vectors. At 24 h post-transfection, cells were incubated with vehicle (**C**), A4 (10^−9^ M), T (10^−9^ M), and each gene-expressing HEK293 cell’s culture medium collected at 2 h after A4 addition. Values of the vehicle were defined as 1. Data represent the mean ± SEM of four independent experiments. Values marked by the different letters are significantly different with each other (*p* < 0.05).

## Data Availability

Data available from the corresponding author upon reasonable request.

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
