# Peer review of "Analyses of Molecular Characteristics and Enzymatic Activities of Ovine HSD17B3"

_animals, 2021, doi:10.3390/ani11102876_

Round 1
Reviewer 1 Report
The work submitted by Islam and colleagues focus in the study of HSD17B3 in sheep (ovis aries). This enzyme plays a key role in the steroidogenic pathway, mainly on males, converting androsterone (A4) into testosterone (T). In this study, Islam et al determined the reading frame of the gene encoding this enzyme. Furthermore, indirect evaluation of the enzymatic activity of the ovine factor, in testosterone and estradiol production, is presented using non-ovine cell lines. Finally, effects of two specific mutations, previously identified in humans, are further indirectly evaluated on its A4 to T conversion capacity.
Although the originality of the manuscript is low, as it largely follows methodology and, in some cases, the descriptions, presented and applied to the human species (Yazawa et al, 2020), there is still some merit in the manuscript by determining HSD17B3 gene sequence in the ovine species and the detrimental effects of the analyzed mutations. Disorder of sex differentiation (DSD) is poorly studied in sheep, although some authors have suggested a higher impact in animal production than believed. Furthermore, the causes for DSD when chromosomal atypia or freemartin syndrome are absent are still poorly understood. Thus, the present work brings new interesting hypothesis to be tested in DSD sheep (mutations in HSD17B3 as a cause for low testosterone production and consequent decreased development of male phenotype) that, after proved, could promote the use of sheep as an animal model for DSD in other species, including humans.
Nevertheless, some concerns should be addressed before considering the present work for publication.
The authors mention that the text was edited on a draft version. However, still some issues regarding grammatical mistakes and typos occur throughout the submitted version of the manuscript that should be corrected and are mentioned in detail ahead.
Authors:
The number of authors appears exaggerated for the current work. The authors are kindly asked to reconsider if all the added co-authors fit to the requirements from Animals guidelines (“Authorship must be limited to those who have contributed substantially to the work reported.”) mainly with regards to funding acquisition and investigation for the present work.
Simple summary/Abstract:
-Simple summary and abstract give a good overview of the work. Nevertheless, the simple summary is quite similar to the abstract and not always simplifies the text to be understoofd by the general public. The authors are incentivized to review this aspect to improve the impact of the manuscript.
-The abstract is above the 200 words limitation determined by the journal guidelines and should be shortened.
-The description of the gene in the abstract, but also in introduction and discussion, as “HSD17B3/Hsd17b3” is rather disturbing in a paper describing a work performed in sheep. Gene nomenclature using only the first letter as capital is used solely for the mouse and rat, that are rarely mentioned in the text. Thus, avoid using the mouse/rat gene nomenclature format throughout the whole manuscript (also introduction and discussion) unless it strictly refers to those two species (this includes also, e.g., HSD17B1 and CYP19A1).
Line 33 and 48 – The authors are suggested to add, e.g., “the”, before “_”, so that the sentence does not start with a character.
Line 34 and 49– replace “to those” for “for those”.
Line 34 – For a higher specificity of the text, replace “of other species” doe “in human”, as is later in the abstract (line 49).
Line 36 – remove “were”
Keywords: The scientific name of the sheep (Ovies aries) should be added to the list.
Introduction
The introduction is well written, with an appropriate extension and addresses several of the topics explored in the manuscript. Nevertheless, a presence of a sentence mentioning clearly that the presence of gene mutations in sheep with DSD or HSD17B3 malfunction were never described would be preferable. Furthermore:
Line 60 – use the plural “adrenals” or “adrenal tissue”
Line 70 – add a comma (,) after “(E1)” or write “of the weak estrogen estrone”
Line 82 – the phrasing/message of the authors is not clear. Do they mean the presence of a sex-determining region Y (SRY-positive)? Please improve the description.
Line 84 – the expression “sexual differentiation of neuron” makes no sense. What do the authors mean with this? In one of the references, neuroendocrine anomalies are mentioned. Is this the aim of this description? This strange phrasing is further repeated in line 311 and should also be corrected/removed.
Materials and methods
All methods are described, mostly by citing other papers. The experimental design appears adequate to the aims of the work, with adequate controls being used. However, some important information is missing to allow the reproducibility of the paper and some details require further attention.
-None of the papers cited in the materials and methods section describe the methodology applied. This happens in RT-PCR, luciferase assay, plasmids construction, western blot, LC-MS/MS and even statistics! Instead, the paper cites a paper that cites a paper that cites a paper… forcing the reader to find 3, 4 or 5 papers before finding the original description of the methodology applied here. This is not only unfair for the authors describing originally the methods, but rather disrespectful for the reader. Thus, cite only papers where the methods were described originally. If necessary, changes to the original methods should further be described.
-Were really TRIzol and TRIsure used in all samples, or some samples with one reagent and other with the other? If so, phrasing needs to be corrected.
-Why was total RNA for testis and adrenals bought and not collected from the slaughterhouse as with the other tissue samples?
-As a matter of good practice when working with total RNA isolations, possible contaminating genomic DNA should be removed with a DNAse treatment. Was this performed? If yes, please describe.
-Did the culture mediums only had FBS added, or other additives (e.g., antibiotics, supplements, etc) were also used? If so, they should be described.
-Furthermore, it would be useful to describe in some detail the culture of cells with conditioned medium. Mainly to allow an easier comprehension of the reader when interpreting the obtained results. This is especially hard to understand with the description later on lines 225-226. Also in the results section, there are some inconsistencies regarding the length of cultures with A4 (24h or 2h). Were there different lengths of pre-treatment? And, if so, why? As far as I understood, medium was conditioned with transfected CV1 cells for 24h before being added for 2h to transfected HEK cells. Is this correct?
-Details regarding western blot are missing. Please describe the dilutions of the antibodies used, as well as secondary antibodies.
-Line 163 – there is an extra space after “InertSep SI” that needs to be removed.
-The standard error of the mean (SEM) is always lower than the descriptive SD, and, thus, is frequently used in scientific publications. SEM is used for the estimation how the mean of the sample is related to the mean of the underlying population. Thus, SEM is an inferring statistical measure and not a descriptive method. Thus, this reviewer would prefer the use of SD instead of SEM in the description of obtained results.
-Also in the statistical analysis section, lines 175-178 appear to repeat exactly the same information of lines 171-172. Please correct.
Results
The results section describes the findings of the manuscript on a clear way. Details on the rational for performed experiments, as well as key interpretation/conclusions, guide the reader through all the experiments done. Still, the following issues should be addressed:
-PCR/Western blot results – According to Animal’s guidelines, “For all Western blot figures, densitometry readings/intensity ratio of each band should be included; the whole Western blot showing all bands and molecular weight markers should be included in the Supplementary Materials.” Please provide this.
-Graphics – despite understanding the limitations on space, using the whole enzyme name (HSD17B1/3, like in Fig.3C)) instead of only B1/3 would make the legend more clear.
-line 184 – remove “that”, as it is not needed in this sentence
-line 185 – the accession number provided is unavailable in the DDBJ website. Please ensure it will be available until the time of publication of the present manuscript.
-line 222 – At a comma after “that”.
-line 225-226 – as mentioned before (in the comments to the materials and methods), the experimental description presented herein is hard to understand. Please rephrase.
-Fig. 2D – Why do the authors think there was a lack of effects with 10-10M of HSD17B3?
-line 256 – The description of HSD17B1 having lower activity than B3 is wrong, as no statistical significance was achieved. However, this reviewer agrees that there is an apparent lower activity of B1.
-Why weren’t T and P4 levels in the medium determined? This would allow a better determination of HSD17B3 activity (and -B1) than the indirect methodology used by the authors.
-line 267 – SEM should be written in capitals to fit with the format used in the rest of the manuscript.
-line 269 – a 3h treatment is mentioned here for the first time. Is this right? And, if so, why the difference in treatment length?
-Regarding the mutations, the description in Fig. 4C does not fit with fig. 4A and 4D (I31V and S289G instead of V31I and G289S). Why is it this way?
-Also, how where the introduced mutations confirmed? Was the material sequenced?
-Where the mutations introduced in the human sequence (4D) further performed in sheep? This would give valuable data for the target species of the present work, the sheep. The addition of this results in the human sequence, although interesting, is out of the scope of the present work. So, if no information is available about these effects in the sheep, that would allow a comparative evaluation with humans, I would suggest removing this.
-line 293 – remove “that”
Discussion/conclusion
Generally speaking, the discussion is poorly written, although some fragments, like the interpretation of HSD17B3 activity on estradiol. Part of it is a repetition of the introduction, with several unrelated topics being widely discussed and some of the focus in the ovine species missing. The authors are advised to reconsider the contents of the discussion.
Some of the questions that should be address are:
-No limitations of the study are described. Where is interesting to observe an effect of testosterone on its own receptor, this is an indirect method of detection. Also, the enzymatic activity of HSD17B3 was described in non-ovine samples. These questions should be addressed.
-The similarity of HSD17B3 sequence to the cow is mentioned, but it ignores the goat as the most similar species.
-Lines 317-319 – how and when was this determined?
-line 320-321 – the system was originally established in Yazawa et al, 2020. This should be at least mentioned.
-line 346 – there is a link/word/expression missing in “ Previous studies ? that murine”
-Remove the information regarding HSD17B5 (lines 354-365), as it has no relevance for the current work.
-please mention that the presence of gene mutations in sheep with DSD or HSD17B3 malfunction were never described.
-references are missing, e.g., in lines 352, 369, 380
Author Response
Please see the attachment. We thank reviewer 1 for the useful indications about our manuscript. We have improved our manuscript according to the suggestions.

Reviewer 2 Report
In this manuscript, the authors have identified the functions of the ovine HSD17B3 they purified. They provide evidence that the HSD17B3 possesses two catalytic activities for the conversions of androstenedione into testosterone, and of estrone into estradiol. Amino acid substituted enzyme activity experiment displays highly conserved gene between ovine and human. While interesting, I would recommend to improve below slightly.
- The introduction is weak. It is unclear why the authors focused on HSD17B3. The sex differentiation, enzyme conversion, and steroidogenesis that HSD17B3 family involves should be elaborated.
- Fig 3 – I could not find the objective of the E1-E2 conversion experiment. I recommend to rewrite the objective of the experiment.
Author Response
Please see the attachment.. We thank Reviewer 2 for the positive comments and useful indications about our manuscript. We have improved our manuscript according to the suggestions.

Round 2
Reviewer 1 Report
In the revised manuscript, Islam and colleagues provided specific answers for each observation or suggestion from the first review, and proceeded with several improvements of the manuscript.
The short abstract was rewritten, with the language being simplified and more accessible to the general public. Furthermore, the main abstract was shortened to comply with the 200 words limit requested by the journal.
Keywords were updated, including now the targeted species in this study. Also, in the introduction, the requested changes to the text were performed, making the overall message more clear.
In the materials and methods, some of the concerns raised, mainly with regards to the acquisition of RNA, lack of DNase treatment, contents of culture medium and western blot methodology were explained by the authors. The addition of a detailed description of cell culture methodology, as requested, made it easier to understand the experimental setup and interpret the results. Furthermore, the citations in this section were revised.
In results section, as required by the journal, pictures of western blots are now provided. Furthermore, in the results and discussion, the requested improvements and suggestions were implemented by the authors. and explanations for the questions made to the authors were provided in the response letter.
Following the first revision process, this reviewer believes that the manuscript should be accepted for publication. Nevertheless, some minor corrections are suggested to improve the quality of the manuscript before the final acceptance:
Lines 38/39 – avoid repetition of the word “species” in the same sentence
Lines 41/43 – similarly, avoid starting two consecutive sentences with the same word “although”.
Line 78 – it seems the authors meant to write “to date” instead of the phrasing present in the manuscript “to data”.
Line 80 – correct “positive”
Line 128 – authors are advised to review the phrasing “supernatants of culture media”
Line 184 - the paper cited in statistics (Yazawa 2021) does not describe any methodology. This should be either replaced or removed before publication.
Line 336 - authors are advised to review the phrasing “possible to useful”
Line 340 – authors are advised to replace “are” for “were”.
Lines 401-404 – the newly added supplemental figure 1, with the picture of the blots, is not described in this section. Also, the original supplemental figure 1 now became supplemental figure 2. This should be corrected.
Line 409 – please remove “please add”
Author Response
We thank reviewer 1 for the useful indications about our revised manuscript. We have improved our manuscript according to the suggestions. Please see the attachment.

This manuscript is a resubmission of an earlier submission. The following is a list of the peer review reports and author responses from that submission.
Round 1
Reviewer 1 Report
This manuscript by Islam and colleagues have cloned, expressed and functional characterized of Ovine 2 HSD17B3 in vitro. This is an interesting topic. However, there are some issue need to be addressed before the publication.
Major comments:
- The writing is really poor. It is really hard to understand. Many grammar and typing errors. This paper have to revise by a native English Speaker.
- Although Figures 3 and 4 characterized the activities of the enzymes, the evidence of protein production of the gene need to be added. Such as western blot or mass spectrometry.
Minor comments:
- Line 37, “uction”?
- Lines 40-41, “we coloned”?
- Line 45, “HSD17B3.Substi-”, space needed.
- Lines 52-53, no key words. The authors really did not respect their work and the journal.
- The abstract is hard to understand.
- Lines 144-155, the presentation of the gene and AA sequences is not easy to ready. Each row could add more information.
- How many replicates have been used in the figures?
Reviewer 2 Report
The manuscript “Cloning, Expression and Functional Analyses of Ovine 2 HSD17B3” by Islam et al. reports that the registered ovine HSD17B3 sequence is incorrect and the role of HSD17B3 with respect to testosterone production.
As the given results were new, the study must be meaningful and may provide an important contribution to comprehension of ovine androgen synthesis in the area of endocrinology and animal husbandry. So, I consider this paper is appropriate for publishing on the Animals. However, the authors need some corrections and addition, and reconsideration in some points.
Major comments:
- The authors cloned ovine HSD17B3 gene and sequenced it and elucidated its role in testosterone production. However, there seems to be no description of which tissue-derived DNA was used and how it was sequenced. Also, the correct sequence claimed by the author should be registered in the DDBJ database before this manuscript is accepted.
- The major 17β-HSDs in the metabolism of testosterone and 5α-dihydrotestosterone are different in humans, mammals, and rodents. What is the expression profile of 17β-HSDs in sheep? Has the enzymatic and/or biological features of other 17β-HSDs such as HSD17B1 or HSD17B5 in sheep that have already been identified? The authors should provide this information for the reader's understanding.
- Line 187-188 (Figure 3B): The authors described ratio of T/A4 activity, but I don't understand how to calculate that. I also don't understand the importance of this result, (e.g., whether it is different from other species.) If it is not very important, it should be removed.
- In Session3.3, the authors evaluate the missense mutation of HSD17B3. To understand the nature of ovine HSD17B3, which is the aim of this paper, it is essentially necessary to prepare the recombinant enzyme and calculate its kinetic parameters. If the author shows that the properties of ovine HSD17B3 are similar to those of the human isozyme, the androgen-binding residues of HSD17B3 should be listed in Fig. 4 and compared among different species.
- About Fig.4C and 4D, since the authors did not use a stable expression system in this experiment, it is difficult to claim that this change in luciferase activity is due to the mutation only. It would be better to ensure the expression level of HSD17B3 by WB or other methods.
Minor comments:
- Line 37: What does “uction of sex steroids” mean?
- Please define at least three keywords.
- The authors should check the text again, as there are a few typos and odd spaces.
- Please define the origin of CV-1 cells.
- Line 144: “Bovidae” is different from “bovine”?
- Line 192: “At 2 D post-transfection” Is this statement correct? The legend in Figure 3 says "At 24 h post-transfection".
- Line 256: in XY DSD of sheep, SNPs of HSD17B3 were identified?
- Line 262-263: The references 25-27 does not seem to match the text. Please also check other places where the literature about human enzymes is cited in the text about ovine enzymes.
Reviewer 3 Report
Dear authors,
I have carefully read your paper entitled “Cloning, Expression and Functional Analyses of Ovine 2 HSD17B3” and unfortunately, I do not think it is suitable for publication on Animals journal, even considering deep revisions. My opinion is that the work carried out does not include cloning or the study of expression, but only functional tests. You can find all my notes in the attached files. I suggest using the data on the functionality of the gene and the mutations present (perhaps analyzing new variants) for the construction of a new paper. These experiments are interesting, and I believe they can be exploited in a new paper. Finally, in the paper there are several concepts and conclusions that are not, in my opinion, correct and therefore strongly collaborate in reducing the general level of the article.
